# HDR-GS: Efficient High Dynamic Range Novel View Synthesis at 1000x Speed via Gaussian Splatting

**Yuanhao Cai** [1], **Zihao Xiao** [1], **Yixun Liang** [2], **Minghan Qin** [3],
**Yulun Zhang** [4,*], **Xiaokang Yang** [4], **Yaoyao Liu** [5], **Alan Yuille** [1]
[1] Johns Hopkins University, [2] HKUST, [3] Tsinghua University,
[4] Shanghai Jiao Tong University, [5] University of Illinois Urbana-Champaign

## Abstract

High dynamic range (HDR) novel view synthesis (NVS) aims to create photorealistic images from novel viewpoints using HDR imaging techniques. The rendered HDR images capture a wider range of brightness levels containing more details of the scene than normal low dynamic range (LDR) images. Existing HDR NVS methods are mainly based on NeRF. They suffer from long training time and slow inference speed. In this paper, we propose a new framework, High Dynamic Range Gaussian Splatting (HDR-GS), which can efficiently render novel HDR views and reconstruct LDR images with a user input exposure time. Specifically, we design a Dual Dynamic Range (DDR) Gaussian point cloud model that uses spherical harmonics to fit HDR color and employs an MLP-based tone-mapper to render LDR color. The HDR and LDR colors are then fed into two Parallel Differentiable Rasterization (PDR) processes to reconstruct HDR and LDR views. To establish the data foundation for the research of 3D Gaussian splatting-based methods in HDR NVS, we recalibrate the camera parameters and compute the initial positions for Gaussian point clouds. Comprehensive experiments show that HDR-GS surpasses the state-of-the-art NeRF-based method by 3.84 and 1.91 dB on LDR and HDR NVS while enjoying $1000\times$ inference speed and only costing 6.3% training time. Code and data are released at https://github.com/caiyuanhao1998/HDR-GS

## 1 Introduction

Compared to normal low dynamic range (LDR) images, high dynamic range (HDR) images capture a broader range of luminance levels to retain the details in dark and bright regions, allowing for more accurate representation of real-world scenes. Novel view synthesis (NVS) aims to produce photo-realistic images of a scene at unobserved viewpoints, given a set of posed images of the same scene. NVS has been widely applied in autonomous driving [1–4], image editing [5–8], digital human [9–12], *etc.* NVS is a challenging topic in computer vision because the limited capacity of camera sensor usually leads to a low dynamic range (from 0 to 255) of luminance in rendered images. This results in the loss of image details in very bright or dark areas, color distortions, and a limited capacity to display subtle gradations in light and shadow that the human eye can normally perceive. Hence, there is a growing demand to render HDR (from 0 to $+\infty$) views for better image quality and visualization performance.

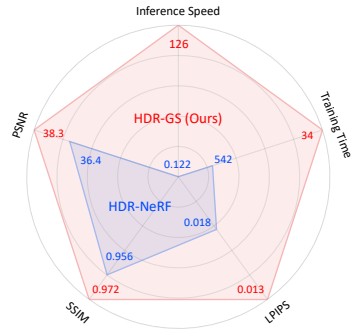

Figure 1: HDR-GS *vs.* HDR-NeRF. Our HDR-GS achieves better PSNR in dB, SSIM, and LPIPS performance with shorter training time in minutes and faster inference speed in fps.

---

[*]Corresponding Author.

38th Conference on Neural Information Processing Systems (NeurIPS 2024).

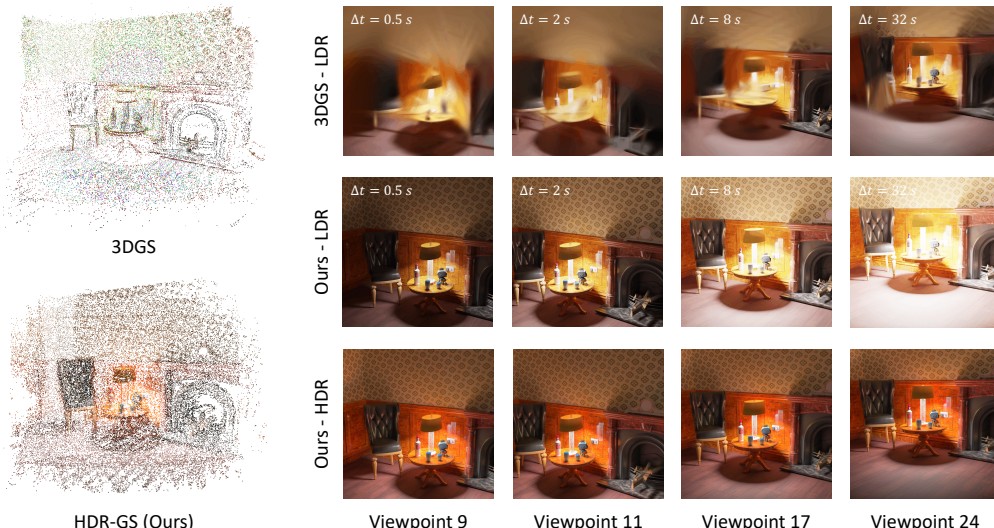

Figure 2: Comparisons of point clouds (left) and rendered views (right) between the original 3DGS [15] (top) and our HDR-GS (bottom). (i) 3DGS [15] renders blurry LDR views when training with images under different exposures. Its point clouds suffer from severe color distortion and can not accurately represent the scene. In addition, 3DGS cannot control the exposure of the rendered images. (ii) Our HDR-GS can not only reconstruct clear HDR images with 3D consistency but also render LDR views with controllable exposure time $\Delta t$.

Existing HDR NVS methods are mainly based on neural radiance fields (NeRF) [13]. However, the ray tracing scheme in NeRF is very time-consuming because it needs to sample many 3D points and then compute their densities and colors for every single ray, severely slowing down the training and inference processes. For instance, the state-of-the-art NeRF-based method HDR-NeRF [14] takes 9 hours to train and 8.2 s to infer an image at the spatial size of 400×400. This limitation impedes the application of NeRF-based algorithms in rendering real-time dynamic scenes.

Recently, 3D Gaussian Splatting (3DGS) [15] has achieved impressive inference speed than NeRF-based methods while yielding comparable results on LDR NVS, which inspires another technical route for HDR NVS. However, directly applying the original 3DGS to HDR imaging may encounter three issues. **Firstly**, the dynamic range of the rendered image is still limited to [0, 255], which severely degrades the visual quality. **Secondly**, training 3DGS with images under different exposures may lead to a non-convergence problem because the spherical harmonics (SH) of 3D Gaussians can not adaptively model the change of exposures. This results in artifacts, blur, and color distortion in the rendered images, as shown in the upper part of Fig. 2. **Thirdly**, 3DGS cannot adapt the exposure level of the synthesized views, which limits its applications, especially in AR/VR, film, and gaming where specific moods and atmospheres are usually evoked by controlling the lighting condition.

To cope with these problems, we propose a novel 3DGS-based method, namely High Dynamic Range Gaussian Splatting (HDR-GS), for 3D HDR imaging. More advanced than the original 3DGS, our HDR-GS can not only render HDR views but also reconstruct LDR images with a controllable exposure time, as depicted in the lower part of Fig. 2. Specifically, we present a Dual Dynamic Range (DDR) Gaussian point cloud model that can jointly model the HDR and LDR colors. We achieve this by using the SH of 3D point clouds to model the HDR color. Then three independent MLPs are employed to model the classical nonparametric camera response function (CRF) calibration process [16] in RGB channels, respectively. By this means, the HDR color of the 3D point is tone-mapped to the corresponding LDR color with the user input exposure time. Subsequently, the HDR and LDR colors are fed into two Parallel Differentiable Rasterization (PDR) processes to render the HDR and LDR images. In addition, we also notice that existing datasets only provide the camera poses in the normalized device coordinate (NDC) system, which is not suitable for 3DGS-based methods. To establish the data foundation for the research of 3DGS-based methods in HDR imaging, we recalibrate the camera parameters and compute SfM [17] points to initialize 3D Gaussians. With the proposed techniques, HDR-GS outperforms state-of-the-art (SOTA) NeRF-based methods by 1.91 dB on the HDR novel view synthesis task while enjoying 1000× inference speed and only requiring 6.3% training time, as shown in Fig. 1. In a nutshell, our contributions can be summarized as:

**(i)** We propose a novel framework, HDR-GS, for 3D HDR imaging. To the best of our knowledge, this is the first attempt to explore the potential of Gaussian splatting in 3D HDR reconstruction.

**(ii)** We present a Dual Dynamic Range Gaussian point cloud model with two Parallel Differentiable Rasterization processes that can render HDR images and LDR views with controllable exposure time.

**(iii)** We establish a data foundation by recalibrating camera parameters and computing initial points for 3DGS-based methods on the multi-view HDR datasets [14]. Experiments show that our HDR-GS dramatically outperforms SOTA methods while enjoying much faster training and inference speed.

## 2 Related Work

**High Dynamic Range Imaging.** Conventional HDR imaging [18] techniques recover HDR images by directly fusing a series of LDR images under different exposure levels at a fixed pose [19] or calibrating the camera response function (CRF) from the LDR images [16, 20]. These traditional methods yield compelling results in static scenes but produce unpleasant ghost artifacts in dynamic scenes. To address this issue, later works [21–25] adopt an optical estimator to detect motion regions in the LDR images and then remove or align these regions in further fusion. With the development of deep learning, convolutional neural networks (CNNs) [26–29] and Transformers [30–33] have been used to learn an implicit mapping function from an LDR image to its HDR counterpart. Yet, these 2D HDR imaging methods lack 3D perception capabilities and are unable to render novel HDR views.

**Neural Radiance Field.** NeRF [13] learns an implicit mapping function from the position of a 3D point and view direction to the point color and volume density. NeRF achieves impressive performance on the NVS task, inspiring many follow-up works to improve its reconstruction quality [34–38] and inference speed [39–45] or expand its application area [14, 46–49]. For example, Huang *et al.* present HDR-NeRF [14] that employs an MLP following the vanilla NeRF to learn an implicit mapping from physical radiance to digital color. Although good results are achieved, these NeRF-based methods suffer from slow training and inference speed due to their time-consuming ray-tracing scheme.

**Gaussian Splatting.** 3DGS [15] explicitly represents a scene by millions of Gaussian point clouds. Its parallelized rasterization in view rendering allows it to enjoy much faster inference speed than NeRF-based methods that suffer from the time-consuming ray-tracing scheme. Thus, 3DGS has been rapidly and widely applied in many areas such as dynamic scene rendering [50–52], SLAM [53–56], inverse rendering [57–59], digital human [60–62], 3D generation [63–65], medical imaging [66], *etc.* However, the illuminance modeled by 3DGS is still limited to a low dynamic range. The potential of 3DGS in HDR imaging still remains under-explored. This work aims to fill this research gap.

## 3 Method

Figure 3 depicts the overall framework of our HDR-GS. To begin with, we use the structure-from-motion (SfM) [17] algorithm to recalibrate the camera parameters of the scene and initialize the Gaussian point clouds, as shown in Fig. 3 (a). Then we propose a Dual Dynamic Range (DDR) Gaussian point cloud model to jointly fit the HDR and LDR colors, as illustrated in Fig. 3 (b). The 3D Gaussians directly use the spherical harmonics (SH) to model the HDR color. Then three independent MLPs are employed to learn the tone-mapping operation in RGB channels. This tone-mapper renders the LDR color from the corresponding HDR color and a controllable exposure time $\Delta t$. Subsequently, the LDR and HDR colors are fed into two Parallel Differentiable Rasterization (PDR) processes to render the HDR and LDR images, as depicted in Fig. 3 (c). In this section, we first introduce the DDR point cloud model, then PDR processes, and finally the initialization and optimization of HDR-GS.

### 3.1 Dual Dynamic Range Gaussian Point Cloud Model

A scene can be represented by a set of our Dual Dynamic Range (DDR) Gaussian point clouds $\mathcal{G}$ as

$$\mathcal{G} = \{G_i(\boldsymbol{\mu}_i, \boldsymbol{\Sigma}_i, \alpha_i, \boldsymbol{c}_i^l, \boldsymbol{c}_i^h, \Delta t, \theta) \mid i = 1, 2, \ldots, N_p\}, \tag{1}$$

where $N_p$ is the number of 3D Gaussians and $G_i$ represents the $i$-th Gaussian. Its center position, covariance, opacity, LDR color, and HDR color are denoted as $\boldsymbol{\mu}_i \in \mathbb{R}^3$, $\boldsymbol{\Sigma}_i \in \mathbb{R}^{3\times3}$, $\alpha_i \in \mathbb{R}$,

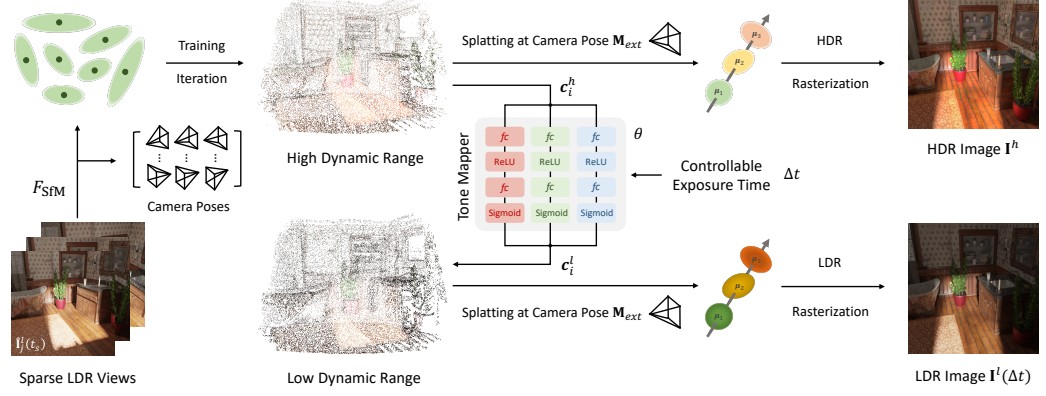

Figure 3: Pipeline of our method. (a) SfM [17] algorithm is used to recalibrate camera parameters and initialize 3D Gaussians. (b) Dual Dynamic Range Gaussian point clouds use spherical harmonics to model the HDR color. Three MLPs are employed to tone-map the LDR color from the HDR color and user input exposure time. (c) The HDR and LDR colors are fed into two Parallel Differentiable Rasterization to render the HDR and LDR views.

$c_i^l \in \mathbb{R}^3$, and $c_i^h \in \mathbb{R}^3$. Besides these attributes, each $G_i$ also contains an exposure time $\Delta t \in \mathbb{R}$ that controls the light intensity of the LDR view and three global-shared MLPs with parameters $\theta$.

$\Sigma_i$ is represented by a rotation matrix $\mathbf{R}_i \in \mathbb{R}^3$ and a scaling matrix $\mathbf{S}_i \in \mathbb{R}^3$ as

$$\Sigma_i = \mathbf{R}_i \mathbf{S}_i \mathbf{S}_i^\top \mathbf{R}_i^\top. \tag{2}$$

$\mu_i$, $\mathbf{R}_i$, $\mathbf{S}_i$, $\alpha_i$, and $\theta$ are learnable parameters. The tone-mapping operation $f_{TM}(\cdot)$ models the camera response function (CRF) that non-linearly maps the HDR color $c_i^h$ into the LDR color $c_i^l$ as

$$c_i^l = f_{TM}(c_i^h \cdot \Delta t), \tag{3}$$

where the exposure time $\Delta t$ can be read from the metadata of photos. We propose to employ MLPs to learn the tone-mapping process. There are two options. The first option is to directly model $f_{TM}(\cdot)$, which may result in the vanishing gradient problem because the multiplication operations may cause numerical overflow or underflow. Besides, the multiplication also leads to the nonlinearity and discontinuity of the input signal of MLPs, which also exacerbates the training instability. The second option is following the traditional non-parametric CRF calibration method of Debevec and Malik [16] that transforms $f_{TM}(\cdot)$ from linear domain to logarithmic domain to enhance the stability of MLP training. We adopt the second option. Specifically, $f_{TM}(\cdot)$ in Eq. (3) is inversed and transformed as

$$\log f_{TM}^{-1}(c_i^l) = \log c_i^h + \log \Delta t, \tag{4}$$

where $\log(\cdot)$ refers to the natural logarithm function and its base is $e = 2.71828\cdots$. Subsequently, we take the inverse function of Eq. (4) on both sides and reformulate it as

$$c_i^l = (\log f_{TM}^{-1})^{-1}(\log c_i^h + \log \Delta t). \tag{5}$$

Then we use three MLPs $\theta$ to model the function $(\log f_{TM}^{-1})^{-1}$ in RGB channels independently because the RGB colors are tone-mapped by different CRFs. For simplicity, we define the mapping function of our tone-mapper $\theta$ as $g_\theta(x) \triangleq (\log f_{TM}^{-1})^{-1}(x)$. Then Eq. (5) is reformulated as

$$c_i^l = g_\theta(\log c_i^h + \log \Delta t), \tag{6}$$

here $\log c_i^h$ is modeled by the spherical harmonics (SH) with a set of coefficients $\mathbf{k} = \{k_l^m | 0 \leq l \leq L, -l \leq m \leq l\} \in \mathbb{R}^{(L+1)^2 \times 3}$. Each $k_l^m \in \mathbb{R}^3$ is a set of three coefficients corresponding to the RGB components. $L$ is the degree of SH. Then $c_i^h$ at the view direction $\mathbf{d} = (\theta, \phi)$ is derived by

$$c_i^h(\mathbf{d}, \mathbf{k}) = \exp\left(\sum_{l=0}^{L} \sum_{m=-l}^{l} k_l^m Y_l^m(\theta, \phi)\right), \tag{7}$$

where $Y_l^m : \mathbb{S}^2 \to \mathbb{R}$ is the SH function that maps 3D points on the sphere to real numbers and $\exp(\cdot)$ represents the the exponential function. By substituting Eq. (7) into Eq. (6), we obtain $\boldsymbol{c}_i^l$ as

$$\boldsymbol{c}_i^l(\mathbf{d}, \mathbf{k}, \Delta t) = g_\theta(\sum_{l=0}^{L} \sum_{m=-l}^{l} k_l^m \, Y_l^m(\theta, \phi) + \log \Delta t + b), \tag{8}$$

here we add a constant bias $b \in \mathbb{R}$ that helps adjust the SH function to better fit the data. The detailed architecture of our MLP-based tone-mapper $\theta$ is shown in Fig. 3 (b). $g_\theta(\cdot)$ equals to the process that the RGB channels of $\log \boldsymbol{c}_h^i$ respectively undergo an independent MLP containing a fully connected ($fc$) layer, a ReLU activation, an $fc$ layer, and a sigmoid activation to produce the LDR color $\boldsymbol{c}_i^l$.

## 3.2 Parallel Differentiable Rasterization

The computed HDR color $\boldsymbol{c}_i^h$ in Eq. (7) and LDR color $\boldsymbol{c}_i^l$ in Eq. (8) of Gaussian point clouds are fed into two parallel differentiable rasterization processes to jointly render the HDR and LDR views, as shown in Fig. 3 (c). The HDR rasterization $F_{\text{HDR}}$ and LDR rasterization $F_{\text{LDR}}$ are represented as

$$\begin{aligned}
\mathbf{I}^h &= F_{\text{HDR}}(\mathbf{M}_{int}, \mathbf{M}_{ext}, \{\boldsymbol{\mu}_i, \boldsymbol{\Sigma}_i, \alpha_i, \boldsymbol{c}_i^h\}_{i=1}^{N_p}), \\
\mathbf{I}^l(\Delta t) &= F_{\text{LDR}}(\mathbf{M}_{int}, \mathbf{M}_{ext}, \{\boldsymbol{\mu}_i, \boldsymbol{\Sigma}_i, \alpha_i, \boldsymbol{c}_i^l(\Delta t)\}_{i=1}^{N_p}),
\end{aligned} \tag{9}$$

where $\mathbf{I}^h$ and $\mathbf{I}^l(\Delta t) \in \mathbb{R}^{H \times W \times 3}$ denote the rendered HDR image and LDR image with the exposure time $\Delta t$, $H$ and $W$ refers to the height and width of the images, $\mathbf{M}_{ext} \in \mathbb{R}^{4 \times 4}$ represents the extrinsic matrix, and $\mathbf{M}_{int} \in \mathbb{R}^{3 \times 4}$ refers to the intrinsic matrix. Please note that we omit $\mathbf{d}$ and $\mathbf{k}$ in $\boldsymbol{c}_i^h$ and $\boldsymbol{c}_i^l$ for simplicity. Then we introduce the details of the parallel rasterization processes. First of all, we derive the possibility value of the $i$-th 3D Gaussian distribution at the point position $\mathbf{x} \in \mathbb{R}^3$ as

$$P(\mathbf{x}|\boldsymbol{\mu}_i, \boldsymbol{\Sigma}_i) = \exp\left(-\frac{1}{2}(\mathbf{x} - \boldsymbol{\mu}_i)^\top \boldsymbol{\Sigma}_i^{-1}(\mathbf{x} - \boldsymbol{\mu}_i)\right). \tag{10}$$

Subsequently, the splatting operation projects the 3D Gaussians to the 2D imaging plane. In this projection process, the center position $\boldsymbol{\mu}_i$ is firstly transferred from the world coordinate system to the camera coordinate system and then projected to the image coordinate system as

$$\widetilde{\mathbf{v}}_i = [\mathbf{v}_i, 1]^\top = \mathbf{M}_{ext} \, \widetilde{\boldsymbol{\mu}}_i = \mathbf{M}_{ext} \, [\boldsymbol{\mu}_i, 1]^\top, \qquad \widetilde{\mathbf{u}}_i = [\mathbf{u}_i, 1]^\top = \mathbf{M}_{int} \, \widetilde{\mathbf{v}}_i = \mathbf{M}_{int} \, [\mathbf{v}_i, 1]^\top, \tag{11}$$

where $\mathbf{u}_i \in \mathbb{R}^2$ and $\mathbf{v}_i \in \mathbb{R}^3$ refer to the image coordinate and camera coordinate of $\boldsymbol{\mu}_i$. $\widetilde{\mathbf{u}}_i$, $\widetilde{\mathbf{v}}_i$, and $\widetilde{\boldsymbol{\mu}}_i$ are the homogeneous versions of $\mathbf{u}_i$, $\mathbf{v}_i$, and $\boldsymbol{\mu}_i$, respectively. The 3D covariance $\boldsymbol{\Sigma}_i$ is also transferred from the world coordinate system to $\boldsymbol{\Sigma}_i^{'} \in \mathbb{R}^{3 \times 3}$ in the camera coordinate system as

$$\boldsymbol{\Sigma}_i^{'} = \mathbf{J}_i \mathbf{W}_i \boldsymbol{\Sigma}_i \mathbf{W}_i^\top \mathbf{J}_i^\top, \tag{12}$$

where $\mathbf{J}_i \in \mathbb{R}^{3 \times 3}$ represents the Jacobian matrix of the affine approximation of the projective transformation $\mathbf{M}_{int}\mathbf{M}_{ext}$. $\mathbf{W}_i \in \mathbb{R}^{3 \times 3}$ is the viewing transformation obtained by taking the first three rows and columns of $\mathbf{M}_{ext}$. Similar to previous works [15, 66–68], the 2D covariance matrix $\boldsymbol{\Sigma}_i^{''} \in \mathbb{R}^{2 \times 2}$ is derived by directly skipping the third row and column of $\boldsymbol{\Sigma}_i^{'}$. Subsequently, the 2D projection is divided into non-overlapping tiles. The 3D Gaussians ($\boldsymbol{\mu}_i$,$\boldsymbol{\Sigma}_i$) are assigned to the tiles where their 2D projections ($\boldsymbol{u}_i$,$\boldsymbol{\Sigma}_i^{''}$) cover. For each tile, the assigned 3D Gaussians are sorted according to the view space depth. Then the RGB value $\mathbf{I}^h(p)$ and $\mathbf{I}^l(p|\Delta t) \in \mathbb{R}^3$ at pixel $p$ is obtained by blending $\mathcal{N}$ ordered points overlapping pixel $p$ in the corresponding tile as

$$\mathbf{I}^h(p) = \sum_{j \in \mathcal{N}} \boldsymbol{c}_j^h \, \sigma_j \prod_{k=1}^{j-1} (1 - \sigma_k), \qquad \mathbf{I}^l(p|\Delta t) = \sum_{j \in \mathcal{N}} \boldsymbol{c}_j^l(\Delta t) \, \sigma_j \prod_{k=1}^{j-1} (1 - \sigma_k), \tag{13}$$

where $\sigma_j = \alpha_j P(\mathbf{x}_j|\boldsymbol{\mu}_j, \boldsymbol{\Sigma}_j)$ and $\mathbf{x}_j$ is the $j$-th intersection 3D point between the ray, which starts from the optical center of the camera and lands at pixel $p$, and the Gaussian point clouds in 3D space. $\boldsymbol{c}_j^h$ and $\boldsymbol{c}_j^l(\Delta t)$ are the HDR color and LDR color with exposure time $\Delta t$ of $\mathbf{x}_j$, respectively.

### 3.3 Initialization and Optimization

An obstacle to the research of 3DGS-based methods in 3D HDR imaging is that the original multi-view HDR datasets [14] only provide the camera poses in the normalized device coordinate (NDC) system. This NDC system is not suitable for 3DGS-based methods for two main reasons. **Firstly**, NDC focuses on describing the positions on the 2D screen after perspective projection. However, 3D Gaussian is an explicit 3D representation. 3DGS requires transforming and projecting Gaussian point clouds in 3D space. **Secondly**, NDC rescales the coordinates into the range [-1, 1] or [0, 1]. The voxel resolution is limited, making it challenging to capture fine details in the scene. **Besides**, the original datasets [14] do not provide SfM [17] point clouds for the initialization of 3DGS.

To address these issues and establish a data foundation for the research of 3DGS-based algorithms in 3D HDR imaging, we use the SfM algorithm [17] to recalibrate the camera parameters and compute the initial positions for 3D Gaussians. The mapping function of SfM $F_{\text{SfM}}$ is summarized as

$$\mathbf{M}_{int}, \{\mathbf{M}_{ext}^j\}_{j=1}^{N_v}, N_p, \{\boldsymbol{\mu}_i\}_{i=1}^{N_p} = F_{\text{SfM}}(\{\hat{\mathbf{I}}_j^l(t_s)\}_{j=1}^{N_v}), \tag{14}$$

where $N_v$ represents the number of viewpoints and $\hat{\mathbf{I}}_j^l(t_s) \in \mathbb{R}^{H \times W \times 3}$ refers to the LDR image at the $j$-th viewpoint with the exposure time $t_s$ in the multi-view HDR datasets [14]. The intrinsic matrix $\mathbf{M}_{int}$ does not change with the viewpoint. Please note that we take the HDR views under the same exposure time as the inputs of SfM algorithms because SfM is based on multi-view feature detection and matching. Changes in exposure conditions may degrade the accuracy of SfM. Then we use the computed $N_p$ and $\{\boldsymbol{\mu}_i\}_{i=1}^{N_p}$ in Eq. (14) to initialize $\mathcal{G}$ in Eq. (1). Other learnable parameters of $\mathcal{G}$ are randomly initialized. The recalibrated camera pose-image data pairs $\{\mathbf{M}_{ext}^j, \hat{\mathbf{I}}_j^l(\Delta t)\}_{j=1}^{N_v}$ in Eq. (14) are used to train our HDR-GS with the weighted sum of $\mathcal{L}_1$ loss and D-SSIM loss as

$$\mathcal{L}_p = \sum_{j=1}^{B} \mathcal{L}_1(\mathbf{I}_j^l(\Delta t_j), \hat{\mathbf{I}}_j^l(\Delta t_j)) + \lambda \cdot \mathcal{L}_{\text{D-SSIM}}(\mathbf{I}_j^l(\Delta t_j), \hat{\mathbf{I}}_j^l(\Delta t_j)), \tag{15}$$

where $B$ is the training batch size and $\lambda$ is a hyperparameter. Similar to HDR-NeRF [14] that uses the ground truth CRF correction coefficient $C_0$ to restrict the HDR color on the synthetic scenes, we also enforce a constraint to the rendered HDR image in the $\mu$-law [14, 23, 69, 70] LDR domain as

$$\mathcal{L}_c = \sum_{j=1}^{B} \left\| \frac{\log(1 + \mu \cdot \text{norm}(\mathbf{I}_j^h))}{\log(1 + \mu)} - \frac{\log(1 + \mu \cdot \text{norm}(\hat{\mathbf{I}}_j^h))}{\log(1 + \mu)} \right\|_2^2, \tag{16}$$

where $\mu$ is the amount of compression. $\text{norm}(\cdot)$ is the min-max normalization. $\mathbf{I}_j^h$ and $\hat{\mathbf{I}}_j^h \in \mathbb{R}^{H \times W \times 3}$ denote the rendered and ground-truth HDR image at the $j$-th viewpoint. The overall training loss is

$$\mathcal{L} = \mathcal{L}_p + \gamma \cdot \mathcal{L}_c, \tag{17}$$

where $\gamma$ is a hyperparameter that controls the relative importance between $\mathcal{L}_p$ and $\mathcal{L}_c$. We do not use $\mathcal{L}_c$ in the real scenes since the ground truth HDR images are not provided in the real datasets. Therefore, we set $\gamma = 0.6$ and 0 in the experiments on the synthetic and real datasets, respectively.

## 4 Experiments

### 4.1 Experimental Settings

**Dataset.** We adopt the multi-view image datasets collected by [14], including 4 real scenes captured by a camera and 8 synthetic scenes created by the software Blender [71]. Images with 5 different exposure time $\{t_1, t_2, t_3, t_4, t_5\}$ are captured at 35 different viewpoints. Following HDR-NeRF [14], images at 18 views with exposure time randomly selected from $\{t_1, t_3, t_5\}$ are used for training while other 17 views at exposure time $\{t_1, t_3, t_5\}$ and $\{t_2, t_4\}$ and HDR images are used for testing.

**Implementation Details.** We implement HDR-GS by PyTorch [72]. The models are trained with the Adam optimizer [73] ($\beta_1 = 0.9$, $\beta_2 = 0.999$, and $\epsilon = 1 \times 10^{-15}$) for $3 \times 10^4$ iterations. The learning rate for point cloud position is initially set to $1.6 \times 10^{-4}$ and exponentially decays to $1.6 \times 10^{-6}$. The learning rates for point feature, opacity, scaling, and rotation are set to $2.5 \times 10^{-3}$, $5 \times 10^{-2}$, $5 \times 10^{-3}$, and $1 \times 10^{-3}$. The learning rate for the tone mapper network is initially set as $5 \times 10^{-4}$ and exponentially decays to $5 \times 10^{-5}$. All experiments are conducted on a single RTX A5000 GPU.

| Method | Training Time (min) | Inference Speed (fps) | LDR-OE ($t_1, t_3, t_5$) PSNR↑ | SSIM↑ | LPIPS↓ | LDR-NE ($t_2, t_4$) PSNR↑ | SSIM↑ | LPIPS↓ | HDR PSNR↑ | SSIM↑ | LPIPS↓ |
|---|---|---|---|---|---|---|---|---|---|---|---|
| NeRF [13] | 405 | 0.190 | 13.97 | 0.555 | 0.376 | 14.51 | 0.522 | 0.428 | — | — | — |
| 3DGS [15] | 38 | 121 | 19.46 | 0.690 | 0.276 | 18.97 | 0.778 | 0.309 | — | — | — |
| NeRF-W [77] | 437 | 0.178 | 29.83 | 0.936 | 0.047 | 29.22 | 0.927 | 0.050 | — | — | — |
| HDR-NeRF [14] | 542 | 0.122 | 39.07 | 0.973 | 0.026 | **37.53** | 0.966 | 0.024 | 36.40 | 0.936 | 0.018 |
| HDR-GS (Ours) | **34** | **126** | **41.10** | **0.982** | **0.011** | 36.33 | **0.977** | **0.016** | **38.31** | **0.972** | **0.013** |

Table 1: Quantitative results on the synthetic datasets. Metrics are averaged over all scenes. LDR-OE denotes the LDR results with exposure time $t_1$, $t_3$, and $t_5$. LDR-NE denotes the LDR results with exposure time $t_2$ and $t_4$. HDR denotes the HDR results. HDR-GS yields the best results on all tracks.

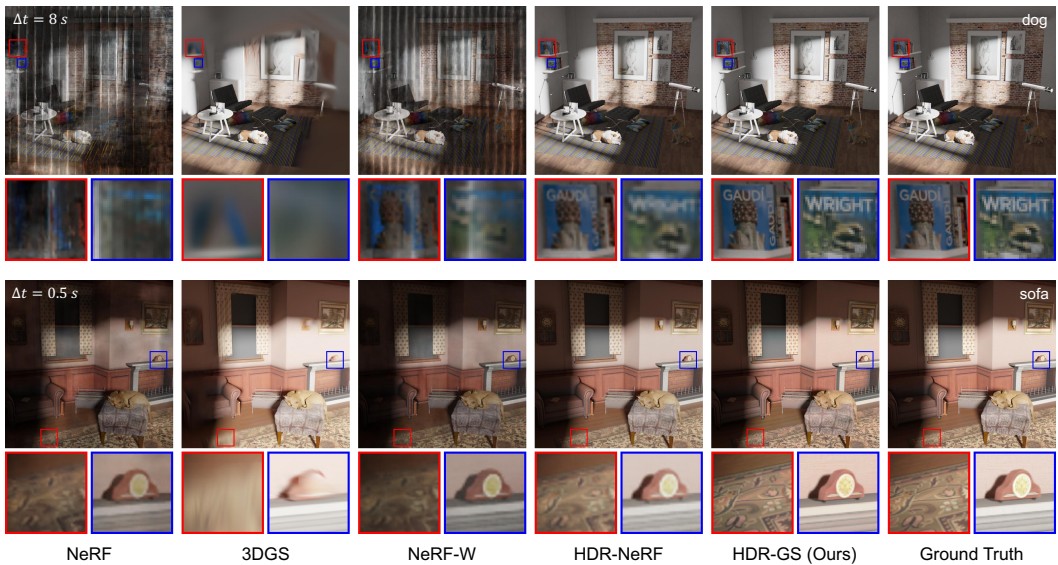

Figure 4: LDR visual comparisons on the synthetic scenes. Previous methods introduce unpleasant black spots or render blurry images. Our method controls the exposure better while reconstructing more detailed structures.

**Evaluation Metrics.** We adopt the peak signal-to-noise ratio, PSNR (higher is better), and structural similarity index measure, SSIM [74] (higher is better), to quantitatively evaluate the objective performance. Learned perceptual image patch similarity, LPIPS [75] (lower is better), is adopted as the perceptual metric. Similar to [14], we also quantitatively evaluate the rendered HDR images in the tone-mapped domain and qualitatively show HDR results tone-mapped by Photomatix pro [76]. In addition, frames per second, fps (higher is faster), is used to measure the model inference speed.

## 4.2 Quantitative Results

**Comparisons on the Synthetic Datasets.** The quantitative results of LDR and HDR NVS on the synthetic datasets are reported in Table 1. We compare our HDR-GS with three NeRF-based methods (NeRF [13], NeRF-W [77], and HDR-NeRF [14]) and the original 3DGS [15]. Table 1 lists the training time, inference speed, PSNR, SSIM, and LPIPS results on LDR-OE, LDR-NE, and HDR. LDR-OE represents the LDR NVS results with exposure time $t_1$, $t_3$, and $t_5$. LDR-NE denotes the LDR NVS results with exposure time $t_2$ and $t_4$. HDR refers to the HDR NVS results. Please note that only HDR-NeRF and HDR-GS can output both LDR and HDR views. Other methods can only render LDR images. Our method outperforms SOTA methods on all tracks except for the PSNR on LDR-NE. **(i)** When compared to the recent best method HDR-NeRF, our HDR-GS outperforms it by 2.03 and 1.91 dB on LDR-OE and HDR tracks while enjoying 1000× faster inference speed and only costing 6.3% training time. **(ii)** When compared to the original 3DGS, our HDR-GS is 21.64 and 17.36 dB higher on LDR-OE and LDR-NE, respectively. Interestingly, HDR-GS is slightly faster

| Method | LDR-OE ($t_1, t_3, t_5$) | | | LDR-NE ($t_2, t_4$) | | |
|---|---|---|---|---|---|---|
| | PSNR↑ | SSIM↑ | LPIPS↓ | PSNR↑ | SSIM↑ | LPIPS↓ |
| NeRF [13] | 14.95 | 0.661 | 0.308 | 14.44 | 0.731 | 0.255 |
| 3DGS [15] | 17.19 | 0.806 | 0.103 | 19.50 | 0.727 | 0.152 |
| NeRF-W [77] | 28.55 | 0.927 | 0.094 | 28.64 | 0.923 | 0.089 |
| HDR-NeRF [14] | 31.63 | 0.948 | 0.069 | 31.43 | 0.943 | 0.069 |
| HDR-GS (Ours) | **35.47** | **0.970** | **0.022** | **31.66** | **0.965** | **0.030** |

Table 2: Quantitative results on the real datasets. Metrics are averaged across all scenes. LDR-OE represents the LDR results with exposure time $t_1$, $t_3$, and $t_5$. LDR-NE denotes the LDR results with exposure time $t_2$ and $t_4$. HDR refers to the HDR results. HDR-GS yields the best results on all tracks.

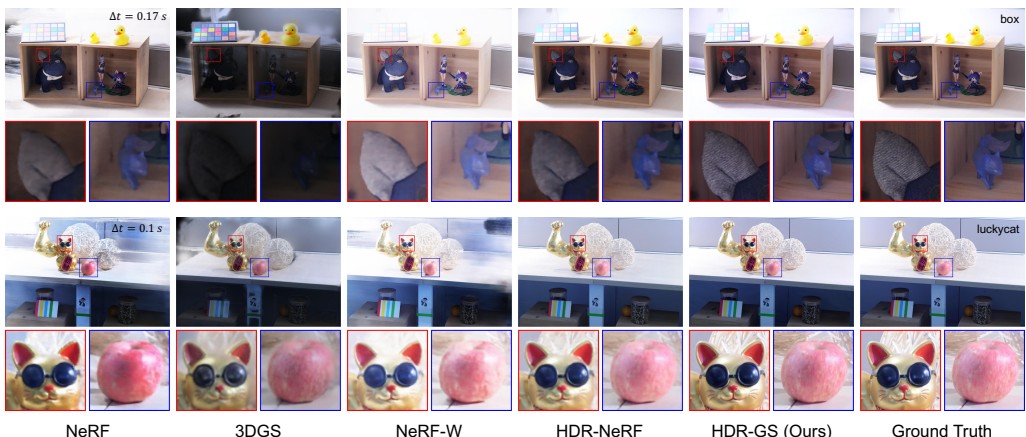

NeRF      3DGS      NeRF-W      HDR-NeRF      HDR-GS (Ours)      Ground Truth

Figure 5: LDR visual comparisons on the real scenes. Previous methods introduce unpleasant black spots or render blurry images. Our method controls the exposure better while reconstructing more detailed structures.

than 3DGS. This is because 3DGS cannot adapt to different exposure levels. It is fragile and hard to converge when training with LDR images under different lighting intensities. The color change of the scene misleads the adaptive density control in 3DGS to split more Gaussian point clouds with distorted color to represent the complex variances in exposure levels, prolonging the training process.

To intuitively show the superiority of our method, Fig. 1 plots a radar chart that features concentric polygons representing the HDR NVS performance across 5 metrics of the SOTA method HDR-NeRF and our HDR-GS. It can be observed that our HDR-GS forms a much larger outermost polygon fully enclosing that of HDR-NeRF, indicating superior performance across all evaluated aspects. These results strongly demonstrate the advantages of our method in effectiveness and model efficiency.

**Comparisons on the Real Datasets.** Table 2 shows the quantitative comparisons on the real datasets. Please note that the real datasets do not provide HDR ground truth for quantitative evaluation. Hence, we only report the LDR NVS results in Table 2. When compared to the recent best method HDR-NeRF, our HDR-GS is 3.84 and 0.23 dB higher in PSNR on LDR-OE and LDR-NE. When compared to the original 3DGS, HDR-GS significantly surpasses it by 18.28 and 12.16 dB on LDR-OE and LDR-NE. These results suggest the outstanding generalization ability and effectiveness of our method.

### 4.3 Qualitative Results

**LDR Novel View Rendering.** The comparisons of LDR novel view rendering with different exposure times on the synthetic (dog and sofa) and real (box and luckycat) scenes are shown in Fig. 4 and 5. It can be observed that NeRF, 3DGS, and NeRF-W fail to control the exposure. They either introduce black stripes and spots, or over-smooth the image, or over-enhance the image while distorting the color. HDR-NeRF can adapt the light intensity but it also produces blurry images. In contrast, our

| Method | Baseline | + Camera Recalibration | + SfM Points | + DDR Model |
|--------|----------|------------------------|--------------|-------------|
| HDR    | –        | –                      | –            | 38.31       |
| LDR-OE | 12.35    | 14.62                  | 19.46        | 41.10       |
| LDR-NE | 11.83    | 14.41                  | 18.97        | 36.33       |

(a) Break-down ablation study towards better performance

| Domain | Linear | Logarithmic |
|--------|--------|-------------|
| HDR    | 26.18  | 38.31       |
| LDR-OE | 29.53  | 41.10       |
| LDR-NE | 27.44  | 36.33       |

(b) Study on the CRF domain

| Exposure | $\{t_3\}$ | $\{t_1, t_5\}$ | $\{t_1, t_3, t_5\}$ | $\{t_1, t_2, t_3, t_4, t_5\}$ |
|----------|-----------|----------------|---------------------|-------------------------------|
| HDR      | 22.86     | 32.06          | 38.31               | 38.50                         |
| LDR-OE   | 23.11     | 34.73          | 41.10               | 41.32                         |
| LDR-NE   | 22.37     | 32.90          | 36.33               | 36.48                         |

(c) Study on the exposure times used in training

| $t_s$ (s) | $t_1 = 0.125$ | $t_2 = 0.25$ | $t_3 = 2$ | $t_4 = 8$ | $t_5 = 32$ |
|-----------|---------------|--------------|-----------|-----------|------------|
| HDR       | 36.88         | 37.90        | 38.16     | 38.31     | 38.05      |
| LDR-OE    | 39.71         | 41.07        | 40.98     | 41.10     | 41.21      |
| LDR-NE    | 35.28         | 35.92        | 36.25     | 36.33     | 36.20      |

(d) Study on the exposure time $t_s$ in recalibration

Table 3: Ablations on the synthetic datasets. The PSNR results on HDR, LDR-OE, and LDR-NE are reported.

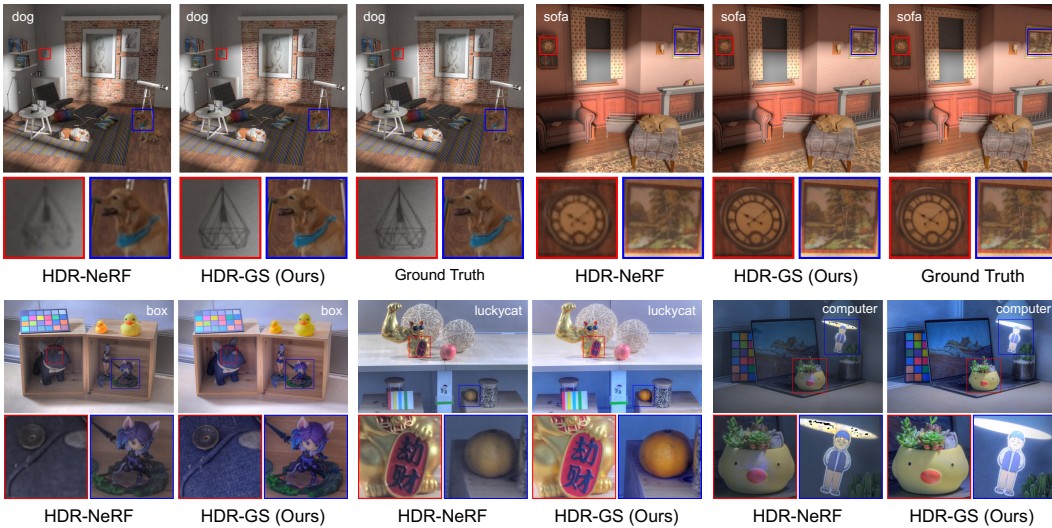

Figure 6: HDR visual comparisons on the synthetic (upper) and real (lower) scenes. Our method can recover the details in both dark and bright regions while suppressing color distortion. Please zoom in for a better view.

HDR-GS can not only control the exposure level of LDR views according to the user input time but also reconstruct clearer images with high-frequency textures and structural contents.

**HDR Novel View Rendering.** The comparisons of HDR novel view synthesis on the synthetic (upper) and real (lower) datasets are depicted in Fig. 6. Please note that only HDR-NeRF and our HDR-GS can reconstruct HDR images. As can be seen, HDR-NeRF yields low-contrast and over-smooth images while sacrificing fine-grained details and introducing undesirable chromatic artifacts and black spots. On the contrary, our HDR-GS can render more perceptually pleasing HDR images with sharper textures and preserve the color and spatial smoothness of homogeneous regions.

### 4.4 Ablation Study

In this section, we adopt the synthetic datasets to conduct ablation study. Table 3 lists the PSNR results averaged across all scenes on the LDR-OE, LDR-NE, and HDR tracks, respectively.

**Break-down Ablation.** We adopt 3DGS [15] trained with the original coordinates (NDC) as the baseline to conduct a break-down ablation on the synthetic datasets. Our goal is to study the effect of each component towards higher performance. The results are reported in Table 3a. **(i)** The baseline model can only render LDR views. It achieves 12.35 and 11.83 dB on LDR-OE and LDR-NE. **(ii)** When using the recalibrated camera poses, the model yields an improvement of 2.27 and 2.58 dB on LDR-OE and LDR-NE because it is liberated from the constraint of the NDC system. **(iii)** When we apply the SfM points for the initialization of 3D Gaussians, the model gains by 4.84 dB and 4.56 dB

because the SfM points provide a general shape of Gaussian point clouds to alleviate the overfitting issues of 3DGS. However, the model still cannot render HDR views nor change the exposure level of the LDR views until now, leading to limited LDR NVS performance. **(iv)** Then we apply our DDR point clouds, the model is enabled to render HDR views with 38.31 dB in PSNR performance. Besides, the model yields 21.64 and 17.36 dB improvements on LDR-OE and LDR-NE because our DDR point clouds allow the model to adapt the lighting intensity with controllable exposure time.

**CRF Domain.** We conduct experiments to compare the effects of modeling CRF in linear domain and logarithmic domain. As shown in Table 3b, when the MLPs $\theta$ directly models $f_{TM}(\cdot)$ in Eq. (3), our method yields poor results of only 26.18, 29.53, and 27.44 dB on HDR, LDR-OE, and LDR-NE. In contrast, when the MLPs $\theta$ models $g_\theta(\cdot)$ in Eq. (6), the performance is 12.13, 11.57, and 8.89 dB higher on HDR, LDR-OE, and LDR-NE. This is because the multiplication in $f_{TM}(\cdot)$ is transferred into the addition in $g_\theta(\cdot)$, which enhances the training stability by suppressing the numerical nonlinearity and discontinuity problems. This evidence verifies our analysis in Sec. 3.1.

**Exposure Time Used for Training.** We conduct experiments in Table 3c to study the effect of the number of exposure times used in training. **(i)** According to the research of Debevec and Malik [16], modeling CRF requires at least two exposures. Thus, when we only use a single exposure $\{t_3\}$, HDR-GS fails to reconstruct HDR views and LDR images with novel exposure time. **(ii)** When two exposures $\{t_1, t_5\}$ are used for training, HDR-GS gains by 9.20, 11.62, and 10.53 dB on HDR, LDR-OE, and LDR-NE. **(iii)** The performance of using three exposures $\{t_1, t_3, t_5\}$ is close to that of using five exposures $\{t_1, t_2, t_3, t_4, t_5\}$. Hence, it is a reasonable choice to use three exposure times.

**Recalibration of Camera Parameters.** In Eq. (14), we use the SfM algorithm to recalibrate the camera parameters and compute the initial positions of 3D Gaussians at the same exposure time $t_s$. Here, we conduct experiments to study the effect of $t_s$ in Table 3d. The performance achieves its maximum value when $t_s = t_4 = 8$ seconds. Therefore, the optimal choice of $t_s$ is $t_4 = 8$ s.

## 5  Limitation and Broader Impact

The main limitation of this work is that the memory usage of 3DGS-based methods is non-trivial and maybe unaffordable to some low-RAM mobile devices. HDR imaging is a very important topic in computational photography. Nowadays, billions of LDR images are captured by mobile phones and cameras. Therefore, how to enhance the quality of these images, adapt the exposure level, and render HDR views is worth studying. Our HDR-GS is capable of reconstructing better HDR and LDR views with controllable exposure time at $1000\times$ speed than SOTA methods, showing great value in practical applications. Until now, 3D HDR imaging techniques have no negative social impact yet. Our proposed HDR-GS does not present any negative foreseeable societal consequences, either.

## 6  Conclusion

This paper focuses on studying the efficiency problem of 3D HDR imaging. We propose the first Gaussian Splatting-based framework, HDR-GS, for HDR novel view synthesis. Our HDR-GS is based on the Dual Dynamic Range Gaussian point clouds that can jointly model HDR color and LDR color with user input exposure time. Then, the HDR and LDR colors are fed into two Parallel Differentiable Rasterization processes to render the HDR and LDR views. To avoid the limitations of NDC system and establish a data foundation for the research of 3DGS-based methods, we recalibrate the camera parameters and compute the initial positions for Gaussian point clouds. Experiments show that our HDR-GS outperforms the SOTA NeRF-based method by 1.91 and 3.84 dB on HDR and LDR novel view rendering, while enjoying $1000\times$ inference speed and requiring only 6.3% training time.

## Acknowledgement

This work was supported by the office of Naval Research with award N000142412696.

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
