# OpenReview forum: "HDR-GS: Efficient High Dynamic Range Novel View Synthesis at 1000x Speed via Gaussian Splatting"
_NeurIPS.cc/2024/Conference — NeurIPS 2024 poster_

### Official Review · Reviewer_VwS8 · 2024-07-06

**Soundness:** 3
**Presentation:** 3
**Contribution:** 2
**Rating:** 6
**Confidence:** 5

**Summary:**

The manuscript presents a novel approach to High Dynamic Range (HDR) novel view synthesis (NVS) by proposing a new framework called High Dynamic Range Gaussian Splatting (HDR-GS). The proposed HDR-GS framework addresses the limitations of existing HDR NVS methods, which are primarily based on Neural Radiance Fields (NeRF) and suffer from long training times and slow inference speeds.

**Strengths:**

Comprehensive experiments demonstrate that HDR-GS outperforms state-of-the-art NeRF-based methods while achieving 1000× faster inference speed and requiring only 6.3% of the training time. The methodology is straightforward and well-organized, and the experimental results are impressive. The paper is logically structured and easy to follow.

**Weaknesses:**

While the methodology is straightforward and the experimental results are strong, the authors could further highlight their innovations and provide detailed explanations of their model design and theoretical underpinnings. This would enhance the manuscript's persuasiveness.

**Questions:**

1)	The authors' method is simple and effective, and the significant speed improvement due to 3D Gaussian Splatting (3D-GS) is not surprising. However, it would be beneficial if the authors could provide an analysis of what specifically leads to the substantial performance difference between HDR-GS and HDR-NeRF.
2)	The authors chose to model HDR color first using the initialized DDR Gaussian point cloud rather than LDR color. It would be helpful if the authors could elaborate on the rationale behind this choice and how it positively impacts the generation of HDR and LDR results.

**Limitations:**

Section 3.2 could be omitted or condensed. The mathematical explanation of 3D-GS rendering and the snowballing process does not seem to be a primary innovation of this paper. Additionally, the emphasis on parallel HDR and LDR rendering and rasterization is somewhat unclear. The authors should clarify why parallel rendering is necessary for their approach, as it seems that similar results could be achieved without parallel rendering. Specifically, the necessity of parallel rendering for the subsequent loss calculation involving 𝐼𝑙 and 𝐼ℎ should be justified.

---

> ### Author Rebuttal · Authors · 2024-08-03
>
> &nbsp;
> ## Response to Reviewer VwS8
> &nbsp;
>
> `Q-1:` Highlight of innovations and explanations of model design and theoretical underpinnings
>
> `A-1:` We propose the first 3DGS-based framework with 1.91 dB improvements and 1000x inference speed for HDR novel view synthesis (NVS). Firstly, we find that directly applying the original 3DGS to HDR NVS does not work well. In Table 3 (a), 3DGS only yields 12.35/11.83 dB on LDR-OE/LDR-NE and cannot render HDR images. To improve the performance, we make innovations in algorithm and data.
>
> (i) Algorithm. The original 3DGS is designed for LDR imaging. If directly enforcing an HDR constraint to the output of 3DGS like Eq.(16), 3DGS only yields a suboptimal result of 30.15 dB on HDR, as shown in the following table. This is because the broader range of brightness and high-frequency details in HDR images are difficult to model by the spherical harmonics with limited order in the original Gaussian point clouds. Plus, 3DGS cannot change the light intensity of the rendered LDR images, thus yielding poor results on LDR.
>
> |Method|LDR-NE|LDR-OE|HDR|
> |:-|:-:|:-:|:-:|
> |3DGS + HDR supervision|15.12|13.47|30.15|
> |HDR-GS|41.10|36.33|38.31|
>
> To address these problems, we design the DDR Gaussian point cloud model based on the tone-mapping theory [16]. Our DDR Gaussian model in Eq.(1) contains more attributes required for HDR NVS, such as HDR color, exposure time, etc. We perform a global tone-mapping operation on all Gaussian point clouds at one time to extract more contextual color information and accelerate the inference speed. Then, we develop the parallel differentiable rasterization to render the LDR and HDR images. As shown in Table 3 (a) of the paper and the above table, our innovation in Gaussian point cloud model leads to significant improvements of 21.64, 17.36, and 8.16 dB on LDR-OE, LDR-NE, and HDR.
>
> (ii) Data. As analyzed in Lines 174 - 181, the normalized device coordinate (NDC) system restricts the representing ability and spatial transformations of 3D Gaussians. Besides, the datasets collected by HDR-NeRF do not provide point clouds for initialization. To address these problems, we recalibrate the camera parameters and compute the initial point clouds as Eq. (14). In Table 3 (a), the data recalibration and point cloud initialization lead to 2.27/2.58 and 4.84/4.56 dB improvements on LDR-OE/LDR-NE, while alleviating the over-fitting issue of 3DGS.
>
> &nbsp;
>
> `Q-2:` Analysis of the performance difference between HDR-GS and HDR-NeRF
>
> `A-2:` We are not sure if the "performance difference" you mention refers to speed or image quality. Thus, we analyze both aspects.
>
> Firstly, two techniques lead to the speed advantage of HDR-GS.
>
> (i) Rendering Scheme. As analyzed in Lines 40 - 45, HDR-NeRF suffers from a time-consuming rendering scheme for LDR and HDR images. It needs to sample many 3D points and then compute their densities and colors for every single ray, severely slowing down the training and inference processes. In contrast, HDR-GS adopts the parallel differentiable rasterization for LDR and HDR rendering. As described in section 3.2, the rasterization computes different tiles divided from the 2D projection in high parallelism on GPU, thus enjoying much faster inference speed.
>
> (ii) Tone-mapping Method. HDR-NeRF conducts tone-mapping in a time-consuming ray-tracing manner. It needs to convert HDR to LDR for all sampled points along every single ray. It can only exploit the limited color information along a single ray for the tone-mapping transformation and requires repeated computation many times. In contrast, HDR-GS performs a global tone-mapping operation that converts the HDR colors of all Gaussian point clouds at one time. This operation can extract more contextual color information for HDR-to-LDR transformation and accelerate the inference speed. The following table compares the effects of the two tone-mapping methods on HDR-GS. Our global tone-mapping method performs better and faster than the ray-tracing tone-mapping of HDR-NeRF.
>
> |Tone-mapping|Train Time (min)|Infer Speed (fps)|LDR-NE|LDR-OE|HDR|
> |:-|:-:|:-:|:-:|:-:|:-:|
> |Ray Tracing (HDR-NeRF)|58|78|39.51|34.68|36.12|
> |Global Infer (Ours)|34|126|41.10|36.33|38.31|
>
> Secondly, although our work is based on 3DGS, directly using 3DGS for HDR NVS yields poor image quality. To improve the quality to surpass HDR-NeRF, we make innovations. Please refer to `A-1` for details.
>
> &nbsp;
>
> `Q-3:` Why modeling the HDR color first?
>
> `A-3:` This is because HDR images capture a broader range of illuminance levels retaining the details in dark and bright regions. HDR images contain information stored in LDR images. Thus, HDR-to-LDR transformation is an easy information compression process. In contrast, the transformation from LDR to HDR is an ill-posed problem because it needs to reconstruct the missing brightness and details that are not captured in LDR images. This process usually involves more complex computations. We do experiments to compare HDR-to-LDR and LDR-to-HDR transformations in the following table. HDR-to-LDR performs better.
>
> |Transformation|LDR-NE|LDR-OE|HDR|
> |:-|:-:|:-:|:-:|
> |LDR-to-HDR|37.65|34.79|29.54|
> |HDR-to-LDR|41.10|36.33|38.31|
>
> &nbsp;
>
> `Q-4:` Condensing section 3.2
>
> `A-4:` We will follow your advice to condense this section
>
> &nbsp;
>
> `Q-5:` Questions about the parallel differentiable rasterization
>
> `A-5:` In fact, parallel differentiable rasterization does not mean parallel computing for HDR and LDR  in one rasterization. We perform HDR rasterization and LDR rasterization, separately. The word "parallel" here has two meanings:
>
> (i) The HDR and LDR rasterization processes query the same attributes and spatial transformations of the same 3D Gaussians in Eq. (10) - (13), except for the color.
>
> (ii) The tiles divided from the 2D projection are computed in parallel on GPU during the rasterization to accelerate the speed, see Line 165 - 169 and Eq. (13).
>
> We will add an explanation in the revision

---

> > ### Comment · Reviewer_VwS8 · 2024-08-13
> >
> > I have read all the reviews and the authors' responses. Most of my concerns have been addressed and I am inclined to keep my previous rating.

---

> > > ### Author Response · Authors · 2024-08-13
> > >
> > > Thanks for keeping your positive view of our paper. We really appreciate it.

---

### Official Review · Reviewer_tLXM · 2024-07-06

**Soundness:** 3
**Presentation:** 4
**Contribution:** 3
**Rating:** 7
**Confidence:** 5

**Summary:**

This paper proposes a 3D Gaussian Splatting-based method, HDR-GS, for the high dynamic range novel view synthesis. To efficiently perform this task, a new Dual Dynamic Range Gaussian point cloud model is presented (in Section 3.1). This point cloud model has more attributes including the HDR color, exposure time, and a global-shared neural network functioning as a tone-mapper. Then in Section 3.2, two Parallel Differentiable Rasterization processes are designed to render the low and high dynamic range colors. Besides, in Section 3.3, the authors recalibrate the camera parameters for the real and synthetic multi-view HDR datasets to make the scene-level data suitable for 3D Gaussian Splatting-based algorithms.

**Strengths:**

+ It is a good attempt to design the first 3DGS-based framework for the task of high dynamic range novel view synthesis. The core idea of assigning more attributes to the Gaussian point cloud model and rendering the high dynamic range and low dynamic range views in the parallel rasterization is novel and cool, which makes the proposed 3D Gaussian model multifunctional and have great practical values in photography, film making, etc.

+ The performance is superior. The running speed of the proposed HDR-GS is more than a thousand time that of the state-of-the-art NeRF-based method, HDR-NeRF. As compared in Table 1 of the main paper, previous NeRF-based methods suffer from the slow inference speed (< 0.2 fps). The proposed HDR-GS can not only infer at a much faster speed of 126 fps (>> 30 fps) but also surpass the SOTA method by large margins. These advantages enable the HDR-GS to capture and measure dynamic scenes, e.g., the camera on the robot, in real time.

+ The writing style is clear and easy to follow. I notice that the authors did not follow the mainstream introduction of the 3D Gaussian Splatting part, which I think is very confusing. Instead, the authors adopted the strategy of summarizing first and then dividing. They first introduced what attributes the Gaussian point cloud model contains, and then gradually introduced its working pipeline. This introduction order can give the readers an overall knowledge of what the Gaussian model is (its attributes) and thus helps the readers better understand the method.

+ The data re-calibration is a bonus. In my opinion, the biggest obstacle to researching the topic of high dynamic range novel view synthesis is the data issue. Although the multi-view synthetic and real datasets are collected, the normalized device coordinates and lack of SfM points for initialization cannot make the 3DGS work because of the severe blur and overfitting problems, especially for the unbounded scene-level reconstruction. How to re-calibrate the data is critical and executing the SfM algorithm is time-consuming.

**Weaknesses:**

There are some minor issues:

- In Line 124 – 131, the authors analyzed the advantages of using log tone-mapper than linear one from the point of view of training stability. Yet, my understanding is that the option of taking the logarithm can shorten the gaps between the training data samples, which makes the originally discrete data samples become more continuous. The processed data samples are easier to fit by neural networks. The exposure time in table 3d is an example. So, adding this analysis can help better explain the motivation of taking the logarithm instead of directly using the linear form.

- In Eq (14), the authors just used part of images of a scene to re-calibrate the camera poses without explanation. My concern is why not use all of the images instead of just using the views under the same exposure time? Did you try that? It is interesting to know and analyze this result.

- It would be better to add some legend or annotation to the teaser figure like the unit of the numerical results, higher is better or lower is better. Because some results are higher is better, e.g., PSNR while some are lower is better such as training time.

- More details of the experimental setup in section 4.1 could be provided to make the implementation clearer. For instance, the authors used the LPIPS as one of the metrics but they did not specify which perceptual network is adopted since this choice may drastically affect the LPIPS score.

**Questions:**

I have two questions:

a) In the paper of HDR-NeRF, the training set and testing sets of real scenes are completely separate and have no intersection according to the implementation details. However, in the official github repository of HDR-NeRF, the training and testing sets for real experiments have intersection, which makes me very confusing. So I want to figure out, in your real experiments, did you separate the training and testing sets or just follow the official code of the HDR-NeRF? I think this is important.

b) I want to know the training stability of the proposed method since I found HDR-NeRF easily collapse and need to train multiple times to make it work on some scenes.

**Limitations:**

Yes, the authors have analyzed the limitations and broader impact of the method in section 4 and 5 of the supplementary pdf.

---

> ### Author Rebuttal · Authors · 2024-08-02
>
> &nbsp;
>
> ## Response to Reviewer tLXM
>
> &nbsp;
>
> `Q-1:` Analysis of logarithmic tone-mapper vs. linear tone-mapper
>
> `A-1:` Thanks for providing a mathematical explanation of the logarithmic tone-mapping operation. We will add this analysis in the revision with acknowledgment.
>
> &nbsp;
>
> `Q-2:` Why using the images under the same exposure of different views to recalibrate the camera parameters instead of using all images?
>
> `A-2:` The recalibration is done by the Structure-from-Motion (SfM) algorithm. SfM needs to detect and then match feature keypoints between different views. Thus, if using all images for calibration, the feature detection and matching might be less accurate due to the change of light intensity at the same position. As a result, the calibrated camera poses may also be inaccurate. We conduct the experiment of using the camera parameters calibrated from all images on the synthetic datasets to train HDR-GS, the model achieves results of only 34.21, 36.19, and 33.75 dB on HDR, LDR-OE, and LDR-NE. Compared to Table 3 (d), these results are 2.67, 3.52, and 1.53 dB lower than the lowest results of single-exposure recalibration with $t_1$ = 0.125 $s$ on HDR, LDR-OE, and LDR-NE.
>
> &nbsp;
>
> `Q-3:` Add some legend or annotation to the teaser figure
>
> `A-3:` Thanks for reminding us. In Figure 1, we report five metrics of HDR novel view synthesis including PSNR in dB, inference speed in fps (frames per second), training time in minutes, SSIM, and LPIPS score. In particular, lower values are better for LPIPS and training time, while higher values are better for the other metrics. We will add a legend to explain the metrics.
>
> &nbsp;
>
> `Q-4:` Which perceptual network is used for the LPIPS score?
>
> `A-4:` Following the same test settings as HDR-NeRF for fair comparison, we use AlexNet [77] as the perceptual network to compute the LPIPS score. We will add more experimental details in Section 4.1.
>
> [77] ImageNet Classification with Deep Convolutional Neural Networks. In NIPS 2012.
>
> &nbsp;
>
> `Q-5:` Are the training sets and testing sets of real scenes completely separate?
>
> `A-5:` Yes, of course. You have a good observation. We also found this mistake in the official repo of HDR-NeRF. We rewrite the data splitting code to make sure there is no overlap between training and testing sets of real scenes. Please check the submitted code, which is consistent with the description of the implementation details in our main paper.
>
> &nbsp;
>
> `Q-6:` The training stability of our HDR-GS
>
> `A-6:` We did not experience model collapse during the training process. Our models were trained once and succeeded. We conduct five repeated experiments on the synthetic datasets. The PSNR results are shown in the following table. The performance fluctuation is within 0.21, 0.16, and 0.09 dB on LDR-OE, LDR-NE, and HDR. These results suggest the robustness and training stability of our HDR-GS.
>
> | Experiment | 1 | 2 | 3 | 4 | 5 | Avg |
> |:-------------|:--:|:--:|:--:|:--:|:--:|:--:|
> | HDR    | 38.31 | 38.23 | 38.34 | 38.22 | 38.29 | 38.28 |
> | LDR-OE | 41.10 | 40.95 | 41.13 | 40.89  | 41.06 | 41.03 |
> | LDR-NE | 36.33 | 36.17 | 36.39 | 36.15 | 36.31 | 36.27 |

---

### Official Review · Reviewer_E3vH · 2024-07-12

**Soundness:** 4
**Presentation:** 4
**Contribution:** 2
**Rating:** 5
**Confidence:** 4

**Summary:**

This paper introduces HDR-GS, a framework designed for efficient rendering of high dynamic range (HDR) novel views. HDR-GS leverages a Dual Dynamic Range (DDR) Gaussian point cloud model that utilizes spherical harmonics for HDR color fitting and an MLP-based tone-mapper for low dynamic range (LDR) color rendering. Given an exposure time, HDR-GS can reconstruct the corresponding LDR image, achieving a form of controllable tone-mapping. The method demonstrates significant improvements over state-of-the-art NeRF-based methods in terms of both speed and image quality.

**Strengths:**

HDR-GS achieves 1000x faster inference speed compared to HDR-NeRF. Its training is also efficient. The results shows significant improvement in image quality.

The framework  is sound. The authors provide a detailed derivation process, demonstrating the motivation and rationale behind the framework design. The tone mapper design enables controllable exposure time for reconstrcting LDR image.

**Weaknesses:**

This method requires taking photos with different exposure settings at each camera position and additional HDR image data in $L_c$ to calculate the loss function. These photos and data are relatively difficult to obtain in practice.

The entire pipeline is quite similar to HDR-NeRF, including the tone mapper MLP. Essentially, the authors replace the NeRF MLP with Gaussian splatting. To adapt HDR-NeRF to Gaussian splatting, the authors propose several key modifications: 1) camera recalibration and point cloud generation; 2) a constant bias $b$ in Equation 8; and 3) using $L_c$ instead of a unit exposure loss $L_u$. However, these modifications are minor and contribute only slightly. Moreover, the poor performance of the baseline model in Table 3a indicates that LDR supervision alone, without GT HDR images, is insufficient for reconstructing the HDR point cloud. This somewhat weakens the novelty. There is not ablation study for the constant bias $b$ in Equation 8.

**Questions:**

My main concern is about the novelty issue mentioned in the Weaknesses section.

Is it possible to apply HDR-GS to data where each viewpoint has a different exposure time?

**Limitations:**

The authors may discuss the difficulty of data acquisition in real-world settings.

---

> ### Author Rebuttal · Authors · 2024-08-04
>
> &nbsp;
> ## Response to Reviewer E3vH
> &nbsp;
>
> `Q-1:` Questions about the training data
>
> `A-1:` (i) Actually, our method only requires an LDR image with a single exposure time $\in$ {$t_1, t_3, t_5$} at each view to train in Eq.(15).
>
> (ii) **As claimed in Lines 198 - 200, we do not use HDR images for supervision $\mathcal{L}_c$ in real scenes, $\gamma = 0$ in Eq.(17).** We only use HDR supervision for synthetic scenes to align the HDR output with the Blender rendering style. This is because the transformation from LDR to HDR is not unique. It is affected by camera parameters, rendering software settings, etc. In our work, synthetic HDR images are synthesized by the software Blender with specific settings and the evaluation metrics for HDR images are hard pixel alignment. Thus, it is reasonable to enforce a constraint on the rendered HDR images. To this end, HDR-NeRF uses the GT camera response function (CRF) correction coefficient $C_{0}$ to rectify the HDR output. But $C_{0}$ is a strong scene-specific prior and not available in real practice because CRF is the target to model, not a given condition. Hence, we use tone-mapped HDR images for supervision instead. For fair comparison, we conduct experiments without using HDR restrictions in the following table. HDR-GS is 14.91 dB higher than HDR-NeRF on HDR.
>
> |Method|HDR-NeRF|HDR-GS|
> |:-|:-:|:-:|
> |HDR|13.35|28.26|
> |LDR-OE|36.48|40.82|
> |LDR-NE|34.47|36.01|
>
> In Figure 6, our method renders more visually pleasant HDR images without HDR supervision than HDR-NeRF in real scenes.
>
> &nbsp;
>
> `Q-2:` Comparison of HDR-GS and HDR-NeRF
>
> `A-2:` Our HDR-GS is different from HDR-NeRF in
>
> (i) Motivation. HDR-NeRF aims to learn the neural HDR radiance fields. It is an implicit 3D representation that lacks geometric information.  In contrast, HDR-GS aims to reconstruct 3D HDR Gaussian point clouds that capture the scene geometry. It is an explicit 3D representation with better controllability and interactivity.
>
> (ii) Technique. (1) In Lines 40 - 45, HDR-NeRF suffers from a time-consuming rendering scheme. It samples many points to compute densities and colors for each ray. In contrast, HDR-GS adopts rasterization to render different tiles in high parallelism on GPU, enjoying much faster speed. (2) The tone-mapping operation of HDR-GS is also fundamentally different from that of HDR-NeRF. HDR-NeRF conducts tone-mapping in a ray-tracing manner. It converts HDR to LDR at all sampled points for every single ray. This ray-tracing tone-mapping only extracts the color information along a single ray and further slows down the inference speed. In contrast, HDR-GS performs a global tone-mapping operation that converts HDR colors of all 3D Gaussians to LDR colors at one time. This operation captures more contextual color information for HDR-to-LDR transformation and accelerates the inference speed. The following table shows the results of using the two tone-mapping methods on HDR-GS. Our global tone-mapping is better and faster.
>
> |Tone-mapping|Train Time (min)|Infer Speed (fps)|LDR-NE|LDR-OE|HDR|
> |:-|:-:|:-:|:-:|:-:|:-:|
> |Ray Tracing (HDR-NeRF)|58|78|39.51|34.68|36.12|
> |Global Infer (Ours)|34|126|41.10|36.33|38.31|
>
> (iii) Data. HDR-NeRF uses the normalized device coordinate (NDC) system. But, as analyzed in Lines 174 - 181, the NDC system restricts the representing ability and transformations of 3D Gaussians. Plus, the datasets collected by HDR-NeRF do not provide point clouds for initialization. To address these problems, we recalibrate the camera parameters and compute the initial point clouds as Eq.(14). See Table 3 (a), our data recalibration and point cloud initialization lead to 2.27/2.58 and 4.84/4.56 dB improvements on LDR-OE/LDR-NE, while alleviating the over-fitting issue.
>
> (iv) Performance. HDR-GS shows significant advantages over HDR-NeRF. HDR-GS surpasses HDR-NeRF by 1.91 dB on HDR in synthetic scenes (Table 1) and 3.84 dB on LDR in real scenes (Table 2), while enjoying 1000x inference speed. In Figure 6, HDR-GS reconstructs clearer HDR details and brightness.
>
> &nbsp;
>
> `Q-3:` Performance of 3DGS trained with GT HDR images
>
> `A-3:` We conduct experiments to compare HDR-GS with 3DGS directly trained with GT HDR images in the following table. HDR-GS outperforms 3DGS by 25.98, 22.86, and 8.16 dB on LDR-NE, LDR-OE, and HDR.
>
> |Method|LDR-NE|LDR-OE|HDR|
> |:-|:-:|:-:|:-:|
> |3DGS|15.12|13.47|30.15|
> |HDR-GS|41.10|36.33|38.31|
>
> We analyze these results:
>
> (i) The suboptimal HDR result of 3DGS stems from that the high-frequency details and broader range of brightness in HDR images are hard to capture by the spherical harmonics (SH) with limited order in original Gaussian point clouds.
>
> (ii) The LDR results of 3DGS are poor because the original 3DGS cannot control the light intensity according to the exposure time.
>
> (iii) Besides, training 3DGS with HDR images is hard in real scenes where HDR images are difficult to obtain. In contrast, HDR-GS only requires LDR images for supervision in practice.
>
> &nbsp;
>
> `Q-4:` Ablation of $b$ in Eq.(8)
>
> `A-4:` We follow your advice to do an ablation of $b$ in the following table.
>
> |Method|LDR-NE|LDR-OE|HDR|
> |:-|:-|:-|:-|
> |w/o $b$|40.72|36.08|38.05|
> |with $b$|41.10|36.33|38.31|
>
> &nbsp;
>
> `Q-5:` Can HDR-GS be applied to data where each view has a different exposure time?
>
> `A-5:` Yes. HDR-GS only requires an LDR image with a single exposure time at each view to train. We do experiments with training views of different exposure times in the following table. HDR-GS performs better and faster.
>
> |Method|Train Time (min)|Infer Speed (fps)|LDR-NE|LDR-OE|HDR|
> |:-|:-:|:-:|:-:|:-:|:-:|
> |HDR-NeRF|551|0.12|37.94|36.21|35.83|
> |HDR-GS|35|123|39.76|37.40|38.05|
>
> &nbsp;
>
> `Q-6:` The difficulty of data acquisition in real world
>
> `A-6:` The real-world data is easy to obtain. In real scenes, HDR-GS only requires a single LDR image at each view to train and no HDR images are required. The exposure time can be easily set in the camera and read from the EXIF files.

---

> > ### Comment · Reviewer_E3vH · 2024-08-10
> >
> > Thank you to the authors for their response; it addressed my concerns regarding HDR supervised training. As for the comparison with HDR-NeRF, I believe most of the differences arise from the necessary adjustments when transitioning between the representation and rendering pipeline of NeRF and 3DGS. These adjustments seem to be natural solutions and do not represent significant novelty. The improvements in rendering quality and speed of HDR-GS also stem from the inherent characteristics of 3DGS. Could the authors point out any technical modifications that are not simply required for adapting the NeRF representation from HDR-NeRF to 3DGS, but that also enhance performance? Or could they explain why some adjustments made during the transition, which could have been done more simply, are superior in their current form?
> >
> > In any case, I appreciate the effort the authors put into this work, which has improved the performance of the results. Considering the rebuttal that has addressed some of my concerns, I am willing to slightly raise the review score.

---

> ### Author Response · Authors · 2024-08-11
> **Discussion with Reviewer E3vH**
>
> Thanks for your reply. We appreciate your recognition of our work and performance.
>
> &nbsp;
>
> Actually, our HDR-GS is not simply adjusting HDR-NeRF to 3DGS. Here are some comparisons in technical details:
>
> &nbsp;
>
> (i) Tone-mapping Operation.
>
> The tone-mapping operation of HDR-NeRF converts the HDR color of the sampled 3D points to LDR color following volume rendering along every single ray. Specifically, the HDR volume rendering is formulated as
>
> $\mathbf{I}^h(\mathbf{r}) = \sum_{i=1}^{N} T_i (1 - \exp(-\rho_i \delta_i)) \mathbf{c}^h_i$,
>
> where $\rho_i$ is the volume density at the $i$-th sampled point. $\mathbf{c}_i^h$ is the HDR color at the $i$-th point. $\mathbf{I}^h(\mathbf{r})$ is the HDR color of the pixel where ray $\mathbf{r}$ lands on. The tone-mapping operation of HDR-NeRF converts $\mathbf{c}^h_i$ to the LDR color $\mathbf{c}^l_i$.
>
> 3DGS also adopts a similar point-based rendering in rasterization. So directly adapting HDR-NeRF to 3DGS should perform the tone-mapping operation following the HDR rasterization. Specifically, in Eq.(13), the point-based rendering in HDR rasterization is
>
> $\mathbf{I}^{h}(p) = \sum_{j \in \mathcal{N}} \mathbf{c}_j^h \sigma_j  \prod _{k=1}^{j-1}(1-\sigma_k)$.
>
> The naive adaptation should perform tone-mapping to convert $\mathbf{c}_j^h$ to $\mathbf{c}_j^l$ along the ray landing on the pixel $p$. However, as compared in the following table, which is copied from the table in `A-2` of our rebuttal for your convenience, we found this adaptation only yields suboptimal results and speed because it can only extract limited color information on a single ray for HDR-to-LDR transformation and needs to compute many times.
>
> |Tone-mapping|Train Time (min)|Infer Speed (fps)|LDR-NE|LDR-OE|HDR|
> |:-|:-:|:-:|:-|:-|:-|
> |Ray Tracing (HDR-NeRF)|58|78|39.51|34.68|36.12|
> |Global Infer (Ours)|34|126|41.10|36.33|38.31|
>
> To address these issues, we design the Dual Dynamic Range (DDR) Gaussian point cloud model that performs a global tone-mapping operation converting the HDR colors of all Gaussian point clouds to LDR at one time to extract more contextual color information and accelerate the inference speed. As shown in the above table, our global tone-mapping leads to improvements of 1.59/1.65/2.19 dB and 48 fps on LDR-NE/LDR-OE/HDR and speed.
>
> &nbsp;
>
> (ii) Data Recalibration.
>
> As analyzed in Lines 174 - 181, HDR-NeRF adopts the normalized device coordinate (NDC) system that rescales the coordinates to the unit cube [-1, 1]$^3$ to help stabilize the training. Plus, the data collected by HDR-NeRF does not provide point clouds for the initialization of Gaussian point clouds.
>
> The naive and straightforward adaptation is to randomly init the positions of Gaussian point clouds within the cube [-1, 1]$^3$ and use the NDC with 2D projections to optimize. However, when we try this naive adaptation, the HDR-GS only achieves poor results of 24.45, 25.31, and 23.08 dB on HDR, LDR-OE, and LDR-NE for two reasons. Firstly, the NDC restricts the representing ability and spatial transformation of Gaussian point clouds. Secondly, training the randomly initialized Gaussian point clouds with few views (only 18) leads to an overfitting issue.
>
> To address these problems, we use the SfM algorithm to recalibrate the camera parameters and compute the initial point clouds. A naive method is to use all images to recalibrate, as mentioned in `Q-2` of our response to reviewer `tLXM`. However, this naive recalibration achieves suboptimal results of only 34.21, 36.19, and 33.75 dB on HDR, LDR-OE, and LDR-NE because the light intensity change degrades the accuracy of feature keypoints detection and matching. Hence, we use the LDR images with the same exposure time to recalibrate in Eq.(14), leading to improvements of 4.10, 4.91, and 2.58 dB on HDR, LDR-OE, and LDR-NE.
>
> &nbsp;
>
> (iii) HDR supervision.
>
> HDR-NeRF uses the ground truth CRF correction coefficient $C_0$ for HDR supervision. The naive adaptation should also use $C_0$. Yet, we found that training with $C_0$ correction is unstable and the model easily collapses. Besides, $C_0$ is unavailable in real scenes where HDR-NeRF naively sets $C_0 = 0.5$. Yet, this inaccurate $C_0$ may cause color distortion or introduce black spots, as shown in Figure 6. Thus, we use Eq.(16) as the HDR supervision for quantitative evaluation to stabilize the training process. When HDR images are not available in practice, we do not use inaccurate $C_0$ to avoid degradation.
>
> &nbsp;
>
> Besides, we resolved the data splitting issue on the real datasets, as mentioned in `Q-5` of our response to reviewer `tLXM`.
>
> &nbsp;
>
> 3DGS is a great work. Yet, directly applying 3DGS or naively adapting HDR-NeRF to 3DGS does not work well. Our work, as the first attempt, proposes an effective method to explore the potential of 3DGS for HDR imaging. We are glad to share our code, model, and recalibrated data with the community.
>
> &nbsp;
>
> Feel free to ask us if you have other questions. Looking forward to your reply.

---

> > ### Author Response · Authors · 2024-08-13
> > **Discussion with Reviewer E3vH**
> >
> > &nbsp;
> >
> > Dear reviewer `E3vH`,
> >
> > &nbsp;
> >
> > Thanks for your time and valuable comments. We appreciate your recognition of our work and effort.
> >
> > Could you please let us know if our response addressed your concerns about the comparison between the simple adaptation of HDR-NeRF for 3DGS and our proposed HDR-GS?
> >
> > We sincerely appreciate your willingness to raise the score. Just a friendly reminder that the scores have not been updated in the openreview system yet.
> >
> > Please feel free to ask us if you have any other questions.
> >
> > We are looking forward to your reply.
> >
> > &nbsp;
> >
> > Best,
> >
> > Authors

---

> > > ### Comment · Reviewer_E3vH · 2024-08-13
> > >
> > > After reading the authors' responses, most of my concerns have been addressed, so I have raised the score.

---

> > > > ### Author Response · Authors · 2024-08-13
> > > >
> > > > Thanks for your consideration. We really appreciate it.

---

### Author Rebuttal · Authors · 2024-08-04

&nbsp;

## General Response to All Reviewers

&nbsp;

Thanks for your time and valuable comments. We really appreciate you for recognizing our framework soundness (`E3vH`,`tLXM`, and `VwS8`), method novelty (`tLXM` and `VwS8`), outstanding performance (`E3vH`,`tLXM`, and `VwS8`), and good writing and presentation (`E3vH`,`tLXM`, and `VwS8`). We have written a separate detailed response to each of you, respectively. We address all the issues raised in detail and clarify a few miscommunications

Our code, models, and recalibrated data will be released to the public.

Feel free to ask us if you have any other questions. Looking forward to discussing with you.

---

### Decision · Program_Chairs · 2024-09-25

**Decision:**

Accept (poster)

**Comment:**

This paper received three positive leaning reviews -- one 5 (borderline accept), one 6 (weak accept), and one 7 (accept).

There was general appreciation for the problem addressed in the paper --- HDR novel view synthesis using 3D GS that allows for considerable speedups both during training and inference. Reviewers also consistently liked the presentation of the paper, quality of the results and were convinced about the effectiveness of the proposed method. There were some concerns about the relatively low technical novelty of the method --- claiming that the paper is a relatively straightforward extension of HDR NeRFs. But on the other hand, there was a consensus that the paper takes a principled approach towards integrating HDR captures into 3DGS framework that provides substantial practical benefits. Given this, an acceptance decision was reached.